# Portable Digital Linear Ion Trap Mass Spectrometer Based on Separate-Region Corona Discharge Ionization Source for On-Site Rapid Detection of Illegal Drugs

**DOI:** 10.3390/molecules27113506

**Published:** 2022-05-30

**Authors:** Lingfeng Li, Tianyi Zhang, Deting Wang, Yunjing Zhang, Xingli He, Xiaozhi Wang, Peng Li

**Affiliations:** 1School of Electronic and Information Engineering, Soochow University, Suzhou 215006, China; lingfengli@suda.edu.cn (L.L.); tyzhangtyzhang1@stu.suda.edu.cn (T.Z.); dtwang0070@suda.edu.cn (D.W.); yjzhang1223@suda.edu.cn (Y.Z.); hexingli@suda.edu.cn (X.H.); 2College of Information Science & Electronic Engineering, Zhejiang University, Hangzhou 310027, China; xw224@zju.edu.cn

**Keywords:** digital linear ion trap, corona discharge, on-site rapid detection, illegal drugs

## Abstract

As narcotic control has become worse in the past decade and the death toll of drug abuse hits a record high, there is an increasing demand for on-site rapid detection of illegal drugs. This work developed a portable digital linear ion trap mass spectrometer based on separate-region corona discharge ionization source to meet this need. A separate design of discharge and reaction regions was adopted with filter air as both carrier gas for the analyte and protection of the corona discharge needle. The linear ion trap was driven by a digital waveform with a low voltage (±100 V) to cover a mass range of 50–500 Da with a unit resolution at a scan rate of 10,000 Da/s. Eighteen representative drugs were analyzed, demonstrating excellent qualitative analysis capability. Tandem mass spectrometry (MS/MS) was also performed by ion isolation and collision-induced dissociation (CID) with air as a buffer gas. With cocaine as an example, over two orders of magnitude dynamic range and 10 pg of detection limit were achieved. A single analysis time of less than 10 s was obtained by comparing the information of characteristic ions and product ions with the built-in database. Analysis of a real-world sample further validated the feasibility of the instrument, with the results benchmarked by GC-MS. The developed system has powerful analytical capability without using consumables including solvent and inert gas, meeting the requirements of on-site rapid detection applications.

## 1. Introduction

According to the 2021 World Drug Report of the United Nations Office on Drugs and Crime (UNODC) [1], the number of people using illegal drugs increased by 22% globally between 2010–2019. In addition, a massive amount of novel psychoactive substances (NPS) such as fentanyl and synthetic cannabinoid are merging. Therefore, for scenarios such as border control, airport, court security, and prison, there is enormous demand for instruments for narcotic detection on the site, which should be sensitive, reliable, rapid, and easy to use. Currently, a number of technologies have been applied in this particular field, such as IMS (ion mobility spectrometry) [2,3], Raman spectrometry [4], FT-IR (Fourier transform infrared spectroscopy) [5], fluorescence techniques [6], and so on [7,8]. However, as the number of target analytes continues to increase, especially NPS, the current technologies and solutions are largely outpaced and found difficult to cope with the growing challenges.

The mass spectrometer, one of the most powerful analytical instruments, features high sensitivity, rapid analysis time, and powerful qualitative analysis capability. Miniaturization of MS for on-site rapid detection is an important direction for the further development of the field. Ion trap mass spectrometer has been considered an optimum choice because of its moderate vacuum requirements, simple structure, and tandem analysis capability [9]. The development of ambient ionization methods without sample preparation or separation such as desorption electrospray ionization (DESI) [10,11], direct analysis in real-time (DART) [11,12], photoionization (PI) [13,14,15], low temperature plasma (LTP) [16,17], paper spray ionization (PSI) [18,19], further paved the way for MS technology from the laboratory use to onsite applications. The feasibility of combining these ambient ionization methods and miniaturized ion trap mass spectrometer for rapid on-site illegal drugs detection has been demonstrated [13,14,20,21,22,23]. However, it normally requires the use of special consumables such as solvent and inert gas, which largely restrain the application.

Corona discharge ionization source has been widely used as a type of atmospheric pressure chemical ionization (APCI) [24,25,26,27,28], which is essentially an ion-molecule reaction in the gas phase. Therefore, the sample introduction method needs to turn the target analyte into the gas-phase. According to the different states of the analyte, various sample introduction methods are required including direct sampling (for gas-phase samples), headspace sampling (for volatile substances in liquid or solid phase), nebulizer (for liquid-phase samples), and thermal desorption (TD, for liquid or solid-phase samples) [29,30,31,32]. However, in most cases, the analyte flow is in direct contact with the discharge needle, which may result in undesired ion fragmentation and further complicate the mass spectrum. Moreover, it also leads to contamination caused by the adsorption of the analyte on the needle tip over time, and eventually failure of discharge. In this work, we proposed a novel construction of septate-region corona discharge ionization source, which is the spatial separation of reactive ion generation and analyte reaction. Meanwhile, filtered air is used as discharge region shielding gas to protect the discharge needle and supplies a stable discharge environment. Controlled reactant ions formed the discharge region and then were introduced into the reaction region by the electrical field and the analytes were ionized. Filtered air is also used for analytes carrier gas, removing the need for an inert gas such as helium or argon as special consumables, which meets the needs of on-site rapid detection. The thermal desorption method was employed to free analyte molecules from the liquid phase or solid phase. Considering the complex background of on-site application, a semi-permeable membrane inlet was introduced to prevent the introduction of unwanted interference. A more detailed design is described in the next section.

Coupling with the novel separate-region corona discharge ionization source, a portable digital linear ion trap mass spectrometer (DLIT-MS) was developed. The sample ions produced from the reaction zone were introduced into the linear ion trap by the discontinuous atmospheric pressure interface (DAPI) method [33]. Although different structures of the linear ion trap mass analyzers have been developed in recent years, such as rectilinear ion trap (RIT) [34], triangular-electrode linear ion trap (TeLIT) [35], and half-round rod electrodes (HreLIT) [36,37], hyperbolic-shaped electrodes ion trap provides most ideal quadrupole electric field and good performance [38] and hence was adopted in our work. For the driving signal, the digital ion trap (DIT) invented by L. Ding et al. in 2002 [39,40] was constructed at a lower voltage (±100 V) which had the advantage of low power consumption and the reduction of the low mass cutoff in tandem mass spectrometry (MS/MS) [40,41]. The qualitative, quantitative, and MS/MS analysis abilities of the developed portable DLIT-MS for eighteen typical illegal drugs were characterized. By comparing information of product ions from the tandem mode with a self-developed database, analysis can be completed within 10 s. Furthermore, analysis of a real-world sample from border custom validated the feasibility of the system for practical use, with the results benchmarked and confirmed by laboratory-based GC-MS. 

## 2. Materials and Methods

### 2.1. Materials and Reagents

HPLC-grade acetonitrile and methanol were purchased from Sigma-Aldrich. The stock solutions of narcotic samples at a concentration of 1000 ng/μL were supplied from Shanghai Yuansi Standard Science and Technology Co., Ltd. (Shanghai, China). The suspected real-world sample, in the form of plant leaves, was provided by border customs of Suzhou city. Molecular sieve was supplied by Sinopharm Chemical Reagent Co., Ltd. (Shanghai, China). Disposable Nomex swab was purchased from Suzhou Chuanche Specialty Materials Co., Ltd. (Suzhou, China). High purity helium (99.999%) for GCMS was purchased from Suzhou Jinhong Gas Co., Ltd. (Suzhou, China).

### 2.2. Sample Preparation and Introduction

The working solutions of narcotic samples were prepared by diluting the stock solution with acetonitrile to target concentrations. Standard liquid solution of analyte was added dropwise onto the Nomex substrate using a pipette with 1 μL aliquot. After evaporation of the solvent in about 10 s, the substrate carrying the analyte was inserted directly into the desorption sampler. For the actual real-world sample, the surface of the sample was scrubbed with the Nomex substrate to collect the analyte, followed by the direct insertion of substrate into the sampler. The Nomex swab was used as disposable to avoid possible interference from residual samples. For GC-MS validation, 1 mg of sample was dissolved in 1.5 mL of methanol for 5 min ultrasonic extraction. The product solution was then filtered with 0.25 μm syringe filter and 1 μL of the resulting solution was analyzed.

### 2.3. GC-MS Characterization and Settings

The measurement was performed on a laboratory-based GC-MS system (Agilent 7000C-7890B) with a DB-5 chromatographic column (30 m × 0.25 mm × 0.25 μm, Agilent Technology Co., Ltd.). The experiment conditions include 280 °C for inlet temperature, 3 min for the solvent delay, 40 °C to 300 °C at 10 °C/min for GC temperature and He with 0.8 mL/min as carrier gas.

### 2.4. Design of the Portable DLIT-MS System

The schematic diagram of the portable DLIT-MS system is shown in Figure 1. The instrument consists of two main parts, a fully integrated sample/ionization assembly including membrane sample inlet, thermal desorption unit and separate-region corona discharge ionization source, and a miniaturized DLIT mass analyzer. 

The sample/ionization assembly comprises a thermal desorption sampler and an ionization chamber with an APCI. The two parts were integrated into a single mechanical structure made of Polyether ether ketone (PEEK). The temperature of the thermal desorption sampler can be controlled from room temperature to 230 °C. The semi-permeable polydimethylsiloxane (PDMS) membrane was used in the sampler which can overcome the problem of contamination introduced by dust, fiber, and airborne particles. The combination of thermal desorption and the use of a membrane can effectively remove the interference of non-volatile substances and reduce the matrix effect. A test paper made of Nomex fiber, which is one of the standard swab materials for commercial explosive trace detector (ETD) instruments [42], was used as the substrate to carry the sample either in the liquid or solid phase. While heated, the analyte passing through the PDMS membrane was carried by the flow of filtered air at 180 mL/min into the ionization chamber. The ionization chamber shown in Appendix A with the dimension of 32 × 30 × 39 mm has a discharge region and a reaction region, which are separated by structure design. The discharge electrode is a piece of a tungsten filament of 80 μm in diameter, which is 2 mm away from the counter electrode with a potential difference of 3000 V. The discharge region is shielded by filtered air to maintain a stable discharge environment over an extended period. The reaction region has an electric field at gradient of 49 V/cm set by the ring electrodes. Shielding gas with a flow rate of 800 mL/min was introduced against the direction of the discharge electric field. The carrier gas and shielding gas were driven by a small diaphragm pump (KVP04-1.1-12, Kamoer Fluildtech (Shanghai, China) Co., ltd.). The product ion is introduced to the vacuum chamber through the DAPI method as described by Cooks’ group [33]. 

A customized DLIT mass analyzer was designed and constructed in this study. The vacuum chamber has dimensions of 73 × 74 × 52 mm and was driven by a 5 L/min diaphragm pump (MVP 003, Pfeiffer Vacuum Inc., Asslar, Germany) and 10 L/s molecular turbo pump (HiPace 10, Pfeiffer Vacuum Inc., Asslar, Germany). The working pressure of the chamber is under 0.01 Pa at stabilized conditions, measured by a Pirani gauge (TPR 280, Pfeiffer Vacuum Inc., Asslar, Germany). The customized linear ion trap we developed comprises two sets of conjugated hyperboloid electrodes and two end cap electrodes. The half-distance between the y (orthogonal direction of ion ejection) electrodes was 5 mm, while the x (direction of ion ejection) electrodes were 0.8 mm stretched and 0.6 mm slotted. The length of the ion trap along z (direction of ion introduction) direction was 40 mm. The structure of the ion trap mass analyzer is shown in Appendix A. The ejected ions from the trap were detected by the combination of a dynode with −6000 V and an electron multiplier (R14747, Hamamatsu Photonics K. K., Hamamatsu, Japan) with −1100 V. 

A set of periodic rectangular RF waves with opposite phases between a high voltage level (100 V) and a low voltage level (−100 V) was applied to the x and y electrodes of the ion trap. The scanning RF frequency was decreased from 1.044 MHz to 0.330 MHz with scan rate of 10,000 Da/s, and an auxiliary dipole AC signal with 1.0~2.5 V was coupled to x electrodes. The AC frequency is fixed at 1/3 of RF frequency, so ions with the mass range of 50–500 Da would be resonantly ejected at β = 2/3. More details about the operation of digital ion trap could be found in other papers [40,41]. Ion isolation and CID can be operated in this DIT system and MS/MS analysis can be performed. 

The integrated sample/ionization assembly, the DAPI system, and the DLIT mass analyzer were incorporated within the instrument housing. All the described hardware and supporting electronics, together with a PC and a power control system, were integrated in a 310 × 310 × 405 mm instrument with a weight of under 15 kg. The whole MS instrument is shown in Appendix A.

## 3. Results and Discussion

### 3.1. Ionization Characteristic of Illegal Drugs in the Ion Source

Eighteen types of illegal drugs were analyzed with the developed instrument, demonstrating its excellent analytical capability. The mass spectra of six representative narcotic samples covering low to high mass range are shown in Figure 2, with the rest to be found in Appendix A of the supporting material. A unit resolution could be achieved, as demonstrated by the characteristic isotopes of ketamine embedded in Figure 2b. From the measured spectra, 16 out of the 18 samples characterized only resulted in [MH]^+^, with the other two also have a small amount of fragment ions, which is similar to that reported in other literature using different soft ionization methods [13,14,20]. Benefit from the design of a separated discharge region, the contamination of the discharge tip can be effectively avoided, resulting in stable conditions for the generation of initial reactant ions. The ionization of analyte molecules is undergoing by the way of charge transfer process in the reaction region. For air as discharge gas as investigated in this work, the initial reactant ion is mainly H(H_2_O)_n_^+^, and the ionization process for analyte M can be described as:H(H_2_O)_n_^+^ +M ↔ MH^+^ (H_2_O)_n_^*^ ↔ MH^+^ + n(H_2_O)(1)

This ionization method with minimal fragmentation is beneficial for the qualitative identification of target analytes from the unknown sample, and is, therefore, a crucial factor for the use of portable DLIT-MS instruments in on-site rapid detection applications. 

Using the last electrode of the reaction region as a faradic plate detector, the measured current due to the initial reactant ion is about 3 nA, much higher than hundreds of pA as in the case of traditional radioactive ionization source [43]. According to the above reaction, a higher density of initial reactant ion moves the reaction to the right and results in a significant amount of analyte ions and hence better sensitivity.

### 3.2. MS/MS and Quantitative Analysis for Illegal Drugs

It is well-known that an ion trap has a limit capacity for ion capture and storage. If there are too many ions in the ion trap, ions with different *m*/*z* will interfere with each other, leading to a compromised analytical result. Especially for the real-world samples, the target analyte is usually presented in a complex background matrix. Tandem mass analysis can significantly reduce this limitation. By isolation of chosen ions in the ion trap, chemical noise can be largely suppressed. Therefore, the tandem MS function for the LDIT-MS was developed in this paper. The full process of tandem MS analysis in the LDIT-MS we developed includes ion introduction, cooling, isolation including digital asymmetric waveform isolation (DAWI) and forward/backward scanning [44], gas introduction, and CID of precursor ion by dipole resonance. This sequence is similar to that described by B. Xue et al. [41]. However, an important alteration is that air was chosen as collision gas instead of helium, which makes the LDIT-MS more practical for the requirement of on-site applications.

Figure 3 shows MS/MS spectra of six typical narcotic samples collected with the developed instrument. The Methamphetamine with the precursor ion of *m*/*z* 150.1 is dissociated into product ions of *m*/*z* 119.1 ([M-CH_4_N]^+^) and 91.1 ([M-C_3_H_8_N]^+^). For ketamine, the precursor ion of *m*/*z* 238.1 mainly produces 220.1 ([M-H_2_O]^+^), 207.1 ([M-CH_4_N]^+^), and 163.0 [M-CH_3_NH-CH_2_OCH_2_]^+^. The morphine of *m*/*z* 286.1 mainly gives the product ions of 268.1([M-H_2_O]^+^), 229.1([M-CH_3_NC_2_H_4_]^+^), 201.1 ([M-C_4_H_7_NO]^+^), and 211.1([M-C_3_H_6_O_2_]^+^). The other samples of the product ions are shown in Figure 3. Spectra of more samples with the product ion information can be found in Appendix A. It could be seen from Figure 3 and Appendix A that the information of product ions with air as CID gas is similar to that obtained from the laboratory-based equipment [45,46]. By dissociating the isolated ions and collecting information on product ions, the analytical capability of the LDIT-MS can be greatly improved.

The tandem MS function offers not only increased capability of identification but also improved ability of quantitative analysis. Although in the application of on-site rapid detection of illegal drugs, quantification of analyte is not the emphasis. It is still valuable to know the dynamic range of the instrument. Taking cocaine as an example, by isolating and dissociating the parent ion of *m*/*z* 304, which is the characteristic ion (MH^+^) for cocaine, the amplitude of the product ion peak at *m*/*z* 182 ([M-C_7_H_6_O_2_]^+^) can be taken as an indicator for quantification. As shown in Figure 4, over two orders of magnitude dynamic range and 10 pg of detection limit could be performed.

### 3.3. Database Construction for Illegal Drugs

For the application of on-site rapid detection, ease of operation, especially by non-professional personnel, is a crucial factor. Therefore, it is essential to have a built-in library for fully automated identification. With the developed instrument, the characteristic ion peaks of all 18 samples have been studied in MS and MS/MS spectra, as shown in Appendix A. A full scan is performed first to find the MS feature of targeted parent ions, followed by CID and MS/MS measurement to check the information of product ions once the parent ion matches. Analysis time of less than 10 s could be achieved in this measuring procedure.

On the other hand, the ionization method mainly causes [MH^+^] which makes it possible to predict the composition of the product ion, laying the foundation for the detection of emerging NPS. The list of narcotics keeps growing and it is practically impossible for a laboratory to have all the samples for study and testing. In case a standard sample is absent, it is possible to use [MH]^+^ as the predicted product ion for the library extension. The MS/MS spectra studied in this work showed that the product ions of the developed instrument are very similar to that of laboratory-based standard equipment [45,46], so it is also possible to expand the library from public data.

### 3.4. Application of the Developed DLIT-MS

A real-world case study has also been included to demonstrate the applicability of the developed instrument. A suspicious unknown item in the form of plant leaves was collected from the border custom of Suzhou city. MS and MS/MS measurements were performed with a simple swab using Nomex substrate, as shown in Figure 5. The main peaks are *m*/*z* 331.2 and 314.2 shown in the mass spectra, matching features of synthetic cannabinoid AB-PINACA and ADB-BUTINACA. Analysis of *m*/*z* 331.2 and 314.2 using the MS/MS method found that the product ion peaks information matched ADB-BUTINACA but not AB-PINACA. Product ions of *m*/*z* 201.1 and 219.1 were further analyzed with MS/MS/MS, resulting in smaller pieces of *m*/*z* 145.0 and 163.0. The product ions pattern found fitted well with the tandem MS analysis result of ADB-BUTINACA reported in the literature [47]. 

To confirm the findings, the sample was also characterized by laboratory-based GC-MS, with its total ion current chromatogram shown in Figure 6b. Corresponding MS spectrum of GC peak with retention of 14.3 min is shown in Figure 6a, which matches well with the standard EI spectrum of ADB-BUTINACA [48] and confirms the accuracy of test results using our portable DLIT-MS instrument. In fact, ADB-BUTINACA is a rather new synthetic cannabinoid which was first found in Europe in 2019 [49]. This case study demonstrated that by following up-to-date literature and analytical information from standard equipment, it is possible to expand the library and detect new analytes even without testing with standard samples.

## 4. Conclusions

A portable digital linear ion trap mass spectrometer based on separate-region corona discharge ionization source was developed for rapid on-site detection of narcotics. The separation of discharge from the reaction region and the use of shield gas were adopted for the design of the ionization source. The developed instrument is fully integrated with compact size and light-weight, and does not require any special consumables such as solvent and inert gas. The investigation of 18 representative narcotic samples with both MS and MS/MS functions demonstrated excellent analytical ability, with over two orders of magnitude dynamic range and 10 pg of the detection limit. In addition, the predictable ionization products and MS/MS product ions pattern enable promising applications, such as forensic analysis and roadside drug screening.

## Figures and Tables

**Figure 1 molecules-27-03506-f001:**
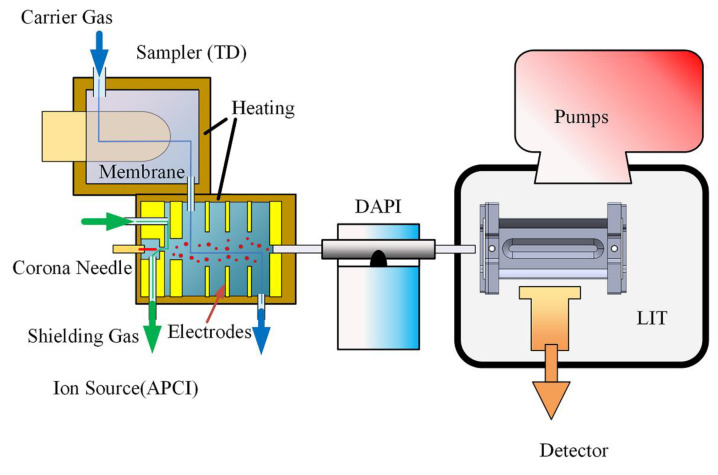
Schematic diagram of the DLIT-MS with the integrated sample/ionization assembly.

**Figure 2 molecules-27-03506-f002:**
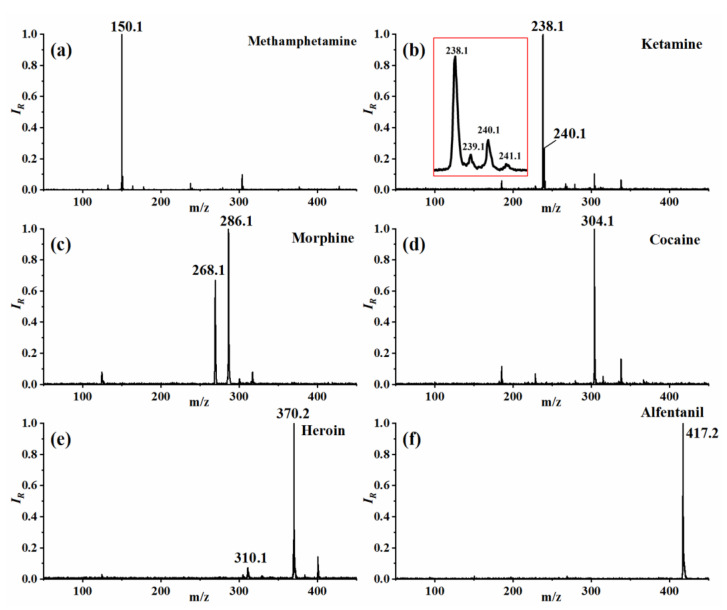
Mass spectra of six representative narcotic samples. (**a**) Methamphetamine; (**b**) Ketamine; (**c**) Morphine; (**d**) Cocaine; (**e**) Heroin; (**f**) Alfentanil. The characteristic isotopes of ketamine were shown in the red box embedded in (**b**).

**Figure 3 molecules-27-03506-f003:**
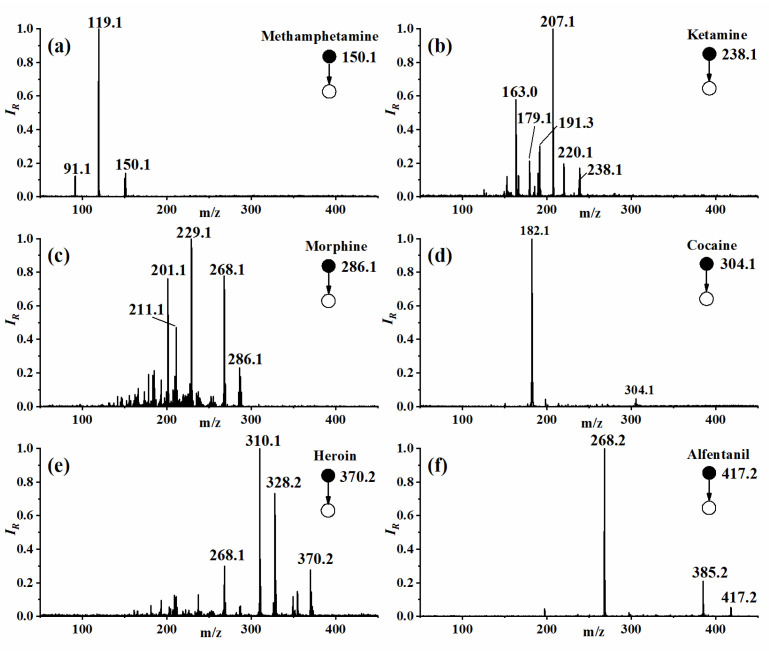
MS/MS spectra of six typical narcotic samples. (**a**) Methamphetamine; (**b**) Ketamine; (**c**) Morphine; (**d**) Cocaine; (**e**) Heroin; (**f**) Alfentanil.

**Figure 4 molecules-27-03506-f004:**
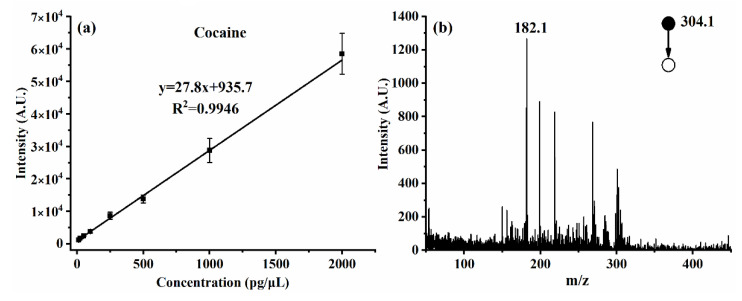
(**a**) Dynamic range of cocaine measured with MS/MS; (**b**) MS/MS spectrum of 10 pg cocaine.

**Figure 5 molecules-27-03506-f005:**
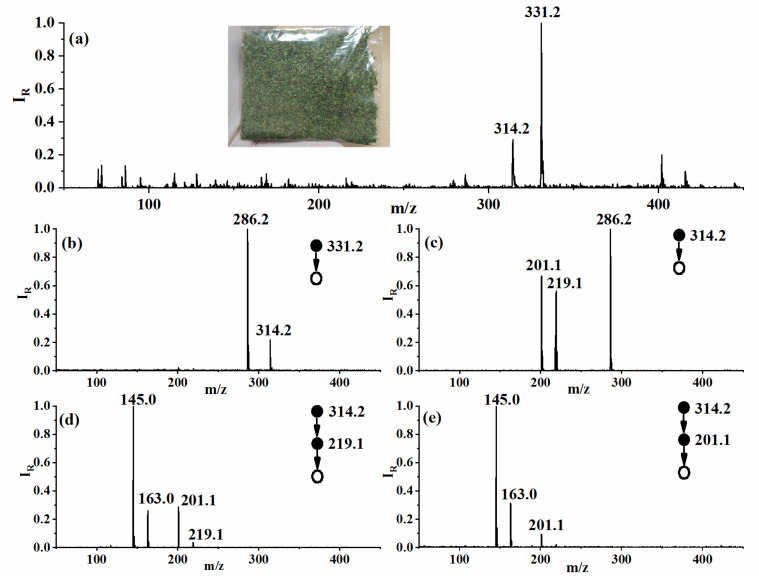
Spectra of a suspicious real-world sample using the DLIT-MS. (**a**) Full scan mass spectrum; (**b**,**c**) MS/MS spectra; (**d**,**e**) MS/MS/MS spectra.

**Figure 6 molecules-27-03506-f006:**
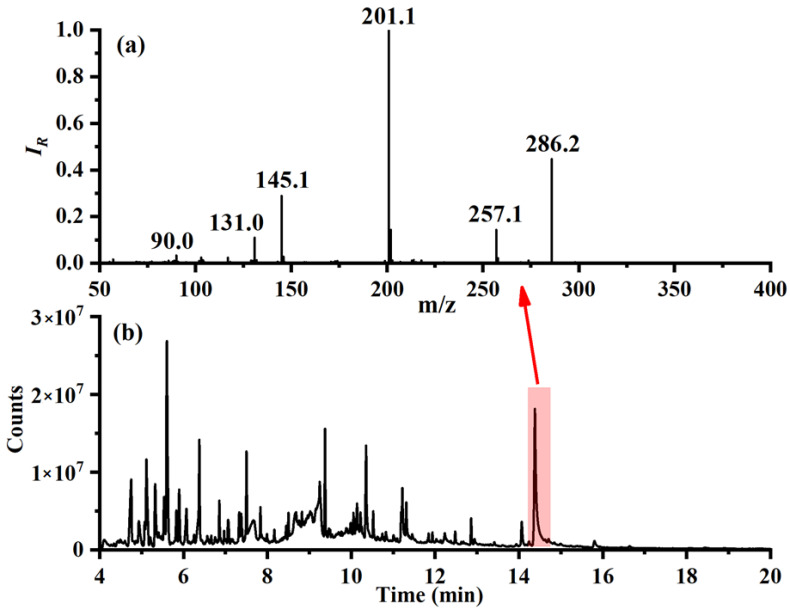
Spectra of the GCMS analysis for the suspicious sample. (**a**) Mass spectrum of GC peak with retention of 14.3 min; (**b**) Total ion current chromatogram.

## Data Availability

The data presented in this work are available in the article and Appendix A.

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
