# Peer review of "Portable Digital Linear Ion Trap Mass Spectrometer Based on Separate-Region Corona Discharge Ionization Source for On-Site Rapid Detection of Illegal Drugs"

_molecules, 2022, doi:10.3390/molecules27113506_

Round 1

Reviewer 1 Report

The authors have presented a manuscript describing the deisng of a new in-house prepared digital linear ion trap MS, using separate-region corona as a source of ions, and a costumized DLIT mass analyzer.

The importante of such an instrument is clearly described by emphasizing the raising issue of illegal drugs and importance of having a portable, "self-suistanable" aparatus. 

In order to be considered for publishing the authors are encouraged to address the following issues:

  1. include a photo of the instrument in Supplementary file. It will be very usefull for readers that have never seen such a portable MS instrument.
  2. Since the method is intended for qualitative analysis a priori, i.e. detecting an illicet drug in a sample, it is crucical to ensure the selectivity of the method and the limit of detection.The MS in general provides excellent selectivity, but the issue of matrix effect should be carefully evaluated and perhaps validated according to the guidelines. Considering that the potential ilegal drug can be incorporated into different matrices in order to descise and hide the substance, it would be worth testing additionaly the influence of the matrix effect.
  3. The authors customized a mass analyzer. Please compare your analyzer to exhisting ones on the market. 
  4. The authors stated that similar fragment information was obtained using air as CID gas to fragmentation patterns obtained using laboratory based equipment. Please include in supplementary the obtained MS/MS spectra of the analytes of interest obtained with the later. 
  5. Figure 4 has the wrong title. It is copy pased from Figure 3.
  6. Please move Figure 5 within the chapter 3.4 and after the text describing the obtained MS/MS spectra of the real world sample. 

Author Response

Point 1: include a photo of the instrument in Supplementary file. It will be very usefull for readers that have never seen such a portable MS instrument.

Response 1: Thank you for your kind suggestion. Photos of the portable MS instrument and ion source have been added in the Supplementary Material.

Point 2: Since the method is intended for qualitative analysis a priori, i.e. detecting an illicet drug in a sample, it is crucical to ensure the selectivity of the method and the limit of detection. The MS in general provides excellent selectivity, but the issue of matrix effect should be carefully evaluated and perhaps validated according to the guidelines. Considering that the potential ilegal drug can be incorporated into different matrices in order to descise and hide the substance, it would be worth testing additionaly the influence of the matrix effect.

Response 2: We totally agree with this argument. Matrix effect is a result of multiple factors including the sampling method, ionization process and mass analysis, and therefore should be dealt with from different aspects in a field deployable instrument which is meant to give results in seconds without length separation process.

In our MS instrument, firstly a semi-permeable polydimethylsiloxane (PDMS) membrane was used in the sampler which can overcome the problem of contamination from dust, fiber, and airborne particles. The combination of thermal desorption and the use of membrane can effectively remove the interference of non-volatile substances. Secondly, the thermal desorption method with carefully selected temperature settings separates different molecules in time according to their own volatility to some extent. Both of these two measures can alleviate the problem of ionization competition. Thirdly, APCI is a charge transfer ionization process based on the proton affinity of molecules which can minimize fragmentation of molecular ions. Fortunately, most of the illegal drug molecules gives high proton affinity which have the competitive advantage in ionization process and mainly produce the ion of [M+H]+. Finally, the mass analyzer provides high resolution, tandem MS function, and good LOD with tens of picograms in our system.

Despite of all above design considerations, we acknowledge that matrix effect might still persist in cases, as there is no limitation how the analyte (illicit drugs in our discussion) exists in field applications. And field deployable instruments, regardless of how carefully designed, may not be able to address all possible situations. Systematical evaluation of field applicability of the developed instrument is still on-going for targeted applications and we think it is beyond the scope of this work which is intended to describe and discuss the design and implementation of the instrument for those possible applications.

Point 3: The authors customized a mass analyzer. Please compare your analyzer to exhisting ones on the market. 

Response 3: Linear ion trap mass analyzer was constructed in our system. Different structures of the linear ion trap mass analyzer were developed in different research group, such as hyperbolic-shaped electrodes ion trap1, rectilinear ion trap (RIT)2, triangular-electrode linear ion trap (TeLIT)3, and half round rod electrodes (HreLIT)4, 5. hyperbolic-shaped electrodes ion trap can provide the closest ideal quadrupole electric field and gives good performance1. The detail of the commercial ion trap with hyperbolic-shaped electrodes MS (LTQ,ThermoFisher) was shown in the work of Jae C. Schwartz etc.1. To miniaturization and simply the structure of the ion trap, a single section device with end cap plates was implemented in our work. Obviously, the performance of our mass analyzer designed for portable MS is not comparable to that of commercial laboratory equipment. However, the performance with the mass range of 50-500 Da, a unit resolution at scan rate of 10000 Da/s, and tandem MS function for our mass analyzer can meet some application of on-site rapid detection (illicit drugs in our discussion). The characteristic isotopes of ketamine measured has embed in the Figure 2(b) to show the performance of resolution. In order to give readers a better understanding of the ion trap structure, the consideration and the references have been added in the manuscript and photo of the ion trap has been attached in the Supplementary Material.

Point 4: The authors stated that similar fragment information was obtained using air as CID gas to fragmentation patterns obtained using laboratory based equipment. Please include in supplementary the obtained MS/MS spectra of the analytes of interest obtained with the later. 

Response 4: We have checked the MS/MS spectra of the eighteen representative drugs with open mass spectral database including MassBank Europe6 and The Human Metabolome Database (hmdb)7. The references of these databases have been added in the revised manuscript.

Point 4: Figure 4 has the wrong title. It is copy pased from Figure 3.

Response 4: Sorry for the mistake and we have revised in the manuscript.

Point 5: Please move Figure 5 within the chapter 3.4 and after the text describing the obtained MS/MS spectra of the real world sample. 

Response 5: Thank you for your kind reminder and we have revised in the manuscript.

Reference:

  1. Schwartz, J. C.; Senko, M. W.; Syka, J. E. P., A two-dimensional quadrupole ion trap mass spectrometer. J. Am. Soc. Mass Spectrom. 2002, 13, 659-669.
  2. Ouyang, Z.; Wu, G.; Song, Y.; Li, H.; Plass, W. R.; Cooks, R. G., Rectilinear Ion Trap:  Concepts, Calculations, and Analytical Performance of a New Mass Analyzer. Anal. Chem. 2004, 76, 4595-4605.
  3. Xiao, Y.; Ding, Z.; Xu, C.; Dai, X.; Fang, X.; Ding, C. F., Novel linear ion trap mass analyzer built with triangular electrodes. Anal. Chem. 2014, 86, 5733-9.
  4. Li, X.; Zhang, X.; Yao, R.; He, Y.; Zhu, Y.; Qian, J., Design and performance evaluation of a linear ion trap mass analyzer featuring half round rod electrodes. J. Am. Soc. Mass Spectrom. 2015, 26, 734-740.
  5. Xu, F.; Wang, W.; Dai, X.; Fang, X.; Ding, C. F., Investigation of the effect of an octopole electric field on a linear ion trap and an asymmetric semi-circular linear ion trap analyzer. Analyst 2021, 146, 6455-6462.
  6. Massbank-High Quality Mass Spectral Database. https://massbank.eu/MassBank/ (accessed 05.15.2022).
  7. HMDB-The Human Metabolme Database. https://hmdb.ca/ (accessed 05.15.2022).

Reviewer 2 Report

The manuscript describes the addition and use of a corona discharge ion source to portable ion trap mass spectrometer, demonstrating the system with a variety of narcotic samples. Although the introduction section mentions a number of well-known ambient sampling/ionization sources, this manuscript should be clearer about the operation of the APCI source and way the sampling is accomplished. Other systems use sampling swabs of various kinds, but the synthetic Nomex fiber, the reason it was chosen compared to other published swab sampling techniques, the way it is cleaned afterward, reusability potential, etc. get little or no description. As the novel addition if the instrument front end is the thrust of this paper, that material must be added. Also, all figures should be made larger than they currently appear in the document to improve readability.

What is the source of airflow? What drives the flow of air through the PDMS membrane and into the APCI chamber, a pressurized/compressed air source pushing through, or a vacuum pulling through? In the case of the former, how would a compressed air source affect system portability? In the case of the latter, is the vacuum from a supplementary pump or from the brief opening oa the DAPI?

I would be instructive to have a photograph of the actual instrumentation in the Supplementary Material, the better to assess the viability of the instrument for field deployment. How big is the APCI chamber compared to the MS?

Is the DAPI connection between the APCI chamber and MS rigid? Other DAPI systems are a solenoid pinching closed a soft flow tube i.e. not rigid. If the transfer is not rigid, how difficult is it to position the two units without breaking the connection, as in a field-deployed setting?

P4 line 141: end-cap electrodes, not end-cup electrodes.

Figure 2: the figure is quite small and difficult to read, and the mass spectra are representative but fairly unremarkable intact ion spectra. You could increase the figure size for readability, but it would be better to take it out and integrate it with Supplementary Figure S1 (or just make it a separate Supplementary figure) because of its simplicity.

P5 line 174 (and elsewhere the phrase is used): “soft ionization” is a common-enough term and not quite deprecated, but it would be better to just eliminate the metaphor and say the technique causes minimal fragmentation.

Figure 3: also quite small as embedded in the text; see comments about Figure 2

Much of Section 3.2 reads like introductory material with results tacked on, like an abstract within the paper written by some else. Revise the section for readability and to integrate with the rest of the manuscript. The results of an MS/MS experiment are properly termed as “product ions”; “fragment ions” are the result of fragmentation of the otherwise-intact molecule upon initial ionization, not MS/MS activation.

Author Response

Point 1: The manuscript describes the addition and use of a corona discharge ion source to portable ion trap mass spectrometer, demonstrating the system with a variety of narcotic samples. Although the introduction section mentions a number of well-known ambient sampling/ionization sources, this manuscript should be clearer about the operation of the APCI source and way the sampling is accomplished. Other systems use sampling swabs of various kinds, but the synthetic Nomex fiber, the reason it was chosen compared to other published swab sampling techniques, the way it is cleaned afterward, reusability potential, etc. get little or no description. As the novel addition if the instrument front end is the thrust of this paper, that material must be added. Also, all figures should be made larger than they currently appear in the document to improve readability.

Response 1: Thanks for your kind suggestion. We have added the common sampling methods of APCI in the introduction section. A reference comparing different materials as swabs has been added in the revised manuscript.

In our study, commercially available disposable synthetic Nomex fiber was used as swabs for sampling, same as other IMS based Explosive Trace Detector (ETD) instruments on the market. We have not investigated alternative material or reusability of the swab, as the thermal desorption sampler was designed purposely for the disposable Nomex fiber (shape & temperature). The information about the Nomex swap paper has been added in the revised manuscript.

The figures have rearranged with larger font in the revised manuscript.

Point 2: What is the source of airflow? What drives the flow of air through the PDMS membrane and into the APCI chamber, a pressurized/compressed air source pushing through, or a vacuum pulling through? In the case of the former, how would a compressed air source affect system portability? In the case of the latter, is the vacuum from a supplementary pump or from the brief opening oa the DAPI?

Response 2: The source of the airflow is a small diaphragm pump with the dimension of Φ24×59.5 mm and it is the way of pushing through. The information of the diaphragm pump has been added in the revised manuscript.  

Point 3: I would be instructive to have a photograph of the actual instrumentation in the Supplementary Material, the better to assess the viability of the instrument for field deployment. How big is the APCI chamber compared to the MS?

Response 3: Thank you for your kind suggestion. The photos of portable MS instrument and the ion source have been added in the Supplementary Material. APCI chamber with the dimension of 32×30×39 mm is much smaller than the whole MS instrument with the dimension of 310×310×405 mm. The information of the APCI chamber dimension has been added in the revised manuscript.

Point 4: Is the DAPI connection between the APCI chamber and MS rigid? Other DAPI systems are a solenoid pinching closed a soft flow tube i.e. not rigid. If the transfer is not rigid, how difficult is it to position the two units without breaking the connection, as in a field-deployed setting?

Response 4: The connection is not rigid. We also used soft flow tube (silicone tube) connecting to stainless steel capillaries with both ends. The relative position of the two units (APCI chamber and vacuum chamber) were fixed by other rigid structural components attaching to both of them.

Point 5: P4 line 141: end-cap electrodes, not end-cup electrodes.

Response 5: Sorry for the mistake and we have revised in the manuscript.

Point 6: Figure 2: the figure is quite small and difficult to read, and the mass spectra are representative but fairly unremarkable intact ion spectra. You could increase the figure size for readability, but it would be better to take it out and integrate it with Supplementary Figure S1 (or just make it a separate Supplementary figure) because of its simplicity.

Response 6: Thank you for your kind suggestion. The figures have rearranged with larger font in the revised manuscript.

Point 7: P5 line 174 (and elsewhere the phrase is used): “soft ionization” is a common-enough term and not quite deprecated, but it would be better to just eliminate the metaphor and say the technique causes minimal fragmentation.

Response 7: Thank you for your kind suggestion. The description has been modified in the revised manuscript.

Point 8: Figure 3: also quite small as embedded in the text; see comments about Figure 2.

Response 8: The figures have rearranged with larger font in the revised manuscript.

Point 9: Much of Section 3.2 reads like introductory material with results tacked on, like an abstract within the paper written by some else. Revise the section for readability and to integrate with the rest of the manuscript. The results of an MS/MS experiment are properly termed as “product ions”; “fragment ions” are the result of fragmentation of the otherwise-intact molecule upon initial ionization, not MS/MS activation.

Response 9: Thank you for your kind advice. We have revised the section 3.2 and made it more fluent. The description of the results of the MS/MS has been modified as “product ions” in the revised manuscript.

Round 2

Reviewer 1 Report

Thank you for the reviesed version of the manuscript and addressing all the reviewers comments. 

Author Response

Thank you for careful review.

Reviewer 2 Report

Now that the instrumentation photographs are included in Supplemental Figure S1, I understand that the APCI desorption cell is incorporated within the portable instrumentation housing. It would be clearer to state that fact explicitly in the text, and to add some labels to Figure S1 to describe the photos.

Author Response

Thank you for your kind suggestion and we have added the description in the manuscript. The Supplemental Figure S1 has been added some labels for the key components.